# The Metabolite Content of the Post-Culture Medium of the Tree Fern *Cyathea delgadii* Sternb. Cell Suspension Cultured in the Presence of 2,4-D and BAP

**DOI:** 10.3390/ijms231911783

**Published:** 2022-10-04

**Authors:** Jan J. Rybczyński, Łukasz Marczak, Maciej Stobiecki, Aleksander Strugała, Anna Mikuła

**Affiliations:** 1Polish Academy of Sciences Botanical Garden-Center for Biological Diversity Conservation in Powsin, 2 Prawdziwka Str., 02-973 Warsaw, Poland; 2Institute of Bioorganic Chemistry, Polish Academy of Sciences, 12/14 Noskowskiego Str, 61-704 Poznań, Poland; 3European Center for Bioinformatics and Genomics, 2 Piotrowo Str., 60-965 Poznań, Poland

**Keywords:** cell suspension, chemical analysis, gas chromatography-mass spectrometry, liquid chromatography-mass spectrometry, plant growth regulators, MS medium

## Abstract

The aim of this study was to demonstrate the metabolic profile of post-culture medium as an expression of cell suspension metabolic activity of the tree fern *Cyathea delgadii* Sternb. The molecular profile of the tree fern’s cell culture has been never described, according to our knowledge. The cell suspension was established using ½ MS medium supplemented with various concentrations of 2,4-D and BAP. The optimal concentrations were 2.0 mg·L^−1^ and 0.2 mg·L^−1^, respectively. The cell suspension initially showed an organized system of cell division and later unorganized cell proliferation. LC-MS and GC-MS were used to identify the chemical composition of the post-culture medium. The LC-MS analysis results suggested that the color of liquid medium could be due to the presence of flavonoid derivatives, as this group of compounds was represented by eight compounds. After GC-MS analysis based on retention indexes and thanks to mass spectra comparison, 130 natural products were recognized, belonging to various classes of primary and secondary metabolites.

## 1. Introduction

The object of our research is the *Cyathea delgadii* Sternb. that in nature is a medium-sized tree fern with a large delicate crow and a very slender trunk [1]. As an experimental plant species, this fern appeared to be very promising due to its excellent morphogenic potential, allowing the formation of somatic embryos without the use of any plant growth regulators, but only by keeping the culture in the dark [2]. This efficient system of plant regeneration has provided new data on somatic embryogenesis studied at different levels of fern morphogenesis, using different methods and tools [3,4]. This scientific information was obtained based on experiments conducted on solidified agar medium. In the current work, we present the results of the response of plant cells that are dispersed in liquid medium supplemented with plant growth regulators, forming a cell suspension culture. According to our experiences with other plant species, this type of culture opens a new area of experimental biology of the tree fern [5].

The suspension cultures of plant cells began in the 1950s [6,7] and were carried out on various plant species; a huge amount of scientific information was gained from that time. The optimization of the culture conditions is still a matter of experiments and should be effective in improving the accumulation of the desire products in cultured cells [8]. At present, they are a source and an unlimited supply of uniform cells and cell aggregates that have a relatively short life cycle and remain undifferentiated. In this type of culture, the plant cells are totally submerged in a liquid medium with forced aeration, which promotes their intensive proliferation. The light regime, sometimes darkness or semidarkness, helps to maintain cell suspension cultures over a long time [9,10]. An agitation of the suspension culture improves uniformity of cell growth, easier control of the culture process and cultivation on a larger scale. The culture is widely used in plant biology as the technical background for work on the somatic hybridization and the transformation of the single cell genome and protoplasts [11,12]. It is also a platform for researchers to investigate plant cell physiology and biochemistry [5]. Cell suspension culture is the most reliable and productive system to generate multiple clones of plants if we are able to deal with its embryogenesity [13]. The high-value secondary metabolites and other substances, like proteins, are of commercial interest [8,14]. Under liquid culture conditions, most of the bioactive substances of plant origin are accumulated intracellularly by cultured cells, thus making their efficient and continuous production very difficult. However, besides that, the chemicals secreted by the plant cells into the surrounding growth medium have also been described [15]. The exudation process involves the secretion of ions, free oxygen and water, enzymes, mucilage, and a diverse array of carbon-containing primary and secondary metabolites. Exudates include low-molecular weight compounds such as amino acids, organic acids, sugars, and phenolics; and high-molecular weight compounds such as polysaccharides and proteins, and their acquisition from in vitro cultures does not affect the biological mass [15]. Only limited information is available about the release of secondary metabolites into the medium, such as alkaloids produced by *Cataranthus roseus* and *Nicotiana rustica* [16,17,18], taxanes by *Taxus* spp. [19,20,21], shikonins by *Lithospermum erythrorhizon* [22] or plumbagin produced by *Plumbago rosea* [23] and genus *Drosera* [24]. In the case of the fern there is a limited number of publications reporting the composition of metabolites in the tissues, or in vitro cell cultures. A comprehensive review on phytochemicals identified in classified fern species and the biological activity of these compounds has been published [25]. Flavonoids, which we also mention in this paper, are widely developed compounds in ferns; these compounds are known for their antioxidative [26] and antimicrobial/antifungal properties. Another class of compounds are pterosins, containing sesquiterpens with 1-indanone skeletons, and showing protective effects on insulin secretion in cells; they also are suspected to have antitumor and anti-inflammatory effects. Ecdysteroids in ferns, their distribution, diversity, biosynthesis and functions are presented in the book: “Working with ferns” [27]. Volatile organic compounds from five French fern species were identified; these natural products belonged to the different classes of natural products [28]. Moreover, many phenolics and antioxidant enzymes were found in the liquid medium of fern cell suspension cultures [29,30]. The limited knowledge of the production of active compounds under in vitro conditions found in cryptogamous species is mainly due to a poor understanding of their biological potential, that could be discovered with the help of current biotechnological tools [31]. Thus, the aim of our experiments was to establish the cell suspension culture of *Cyathea delgadii*, and to describe the chemical composition of the post-culture medium as an expression of intensive cell metabolism occurring during a two-week long subculture.

## 2. Results

Metabolite profiling of the *C. delgadii* post-culture media was performed on 3 years old cell suspension which was subcultured each second week with 3 g of tissue suspended in 80 mL of fresh ½ MS medium supplemented with 2.0 mg·L^−1^ of 2,4-D and 0.2 mg·L^−1^ of BAP, and 20 g·L^−1^ sucrose. Both cell aggregates and post-culture liquid medium were yellow colored (Figure 1). The collected medium was subjected for metabolites profiling with GC-MS and LC-MS methods.

### 2.1. Description of GC-MS Analysis Results

The post-culture liquid medium, following 2 weeks of culture, was analyzed with the GC-MS method for the presence of primary and secondary metabolites. Using this approach, 130 natural products were identified (Table 1 and Appendix A) based on their respective retention times and the registered mass spectra of the trimethylsilyl derivatives of the components of *C. delgadii* culture medium. The identified compounds have been divided into 11 classes/groups of metabolites. Among these groups were: alcohols (4), amines (4), amino acids (10), fatty acids (12), nucleic acids components (5), organic acids (29), phenolics (6), phosphate derivatives (6), sugars/carbohydrates (27), sugar alcohols (15) and 12 other compounds. The most numerous groups were sugars, which could be divided into monosaccharides containing six or four carbons, and few disaccharides. The mono- and di-saccharides were identified with the retention time ranging from 16.52 to 24.12 (Table 1, Figure 2). Other metabolites were observed in different retention times throughout analysis, but accordingly to sugars, specific time regions for their presence could be distinguished as well. The relative amounts of the compounds identified in post-culture medium differed substantially (Figure 2).

### 2.2. Description of LC-MS Analysis Results

The LC-MS analysis was applied to determine the chemical background of the yellow coloration in post-culture medium. On the basis of retention times and CID (collision induced dissociation), the MS/MS spectra or exact *m*/*z* values of ions registered both in positive and negative mode, eight phenolic/flavonoid derivatives were unambiguously identified (Table 2; Figure 3 and Figure 4).

## 3. Discussion

The metabolic profile of post-culture liquid medium derived after a two week long passage of the *C. delgadii* cell suspensions were analyzed with GC-MS and LC-MS.

GC-MS untargeted metabolic profiling of cell culture medium permitted to identify 130 natural products belonging to 11 classes. Four of them (myo-inosytol, sucrose, nicotinic acid, glycine) were included as elements of the MS medium solution to maintain the life cycle of cultured cells [5]. A few compounds (only five) presenting nucleic acid components [36] was observed. It can be the result of cell activity that produce proteins involved in the metabolism of growing cell and later metabolism connected with the production of other natural products [5]. Another group of compounds with a similar number of components were phosphate derivatives (six). Phosphoenolpyruvate and glucose phosphates are known for their role in the glycolysis pathway, which leads to the synthesis of pyruvate from glucose for energy production [37]. Inositol phosphate is an important compound playing a crucial role in cell growth and apoptosis [38].

The most numerous classes of identified compounds were organic acids (29) and sugars (27). It is well known that organic acids are produced by mitochondria, from the catabolism of amino acids, and stored in the vacuoles. They are intermediates in numerous fat and carbohydrate degradation pathways. In an intact plant, a stream of organic acid transfers ions from the roots to the leaves, and in the opposite direction there is a flux of exudation through the roots to the soil [39]. In the case of cell aggregates maintained in liquid medium, the model described above does not work at all. Very intensive metabolism of mitotically active aggregate cells requires energy because of the intensive work of the mitochondria producing organic acids, which are not only stored in the vacuole, but also exuded into the surrounding medium.

Hexoses, pentoses, di- and trisaccharides were also identified in suspension culture of *C. delgadii*. The sugars and the sugar alcohols exuded to the medium play a very important role in the tree fern cultures. They retain water in cells and could be used for the synthesis of phenolic compounds. They are components of cell walls, and also participate together with sugar alcohols in the synthesis of different important, from the physiological point of view, oligosaccharides (building blocks of cell walls), or metabolite glycosides sequestered in the cell vacuoles, or participating in different organelles in the scavenging of reactive oxygen species. The medium contained initially 20 g·L^−1^ of sucrose and no organic acids; thus, we speculate that it might indicate that a high metabolism of sugars is taking place, with sucrose putatively converted into monosaccharides. Three of these, namely the free forms of mannose, glucose and fructose, but especially mannose, may confirm their critical role in the cell wall formation during cell proliferation in the leaves of the fern *Adiantum raddianum*, where it provided evidence of a mannose-rich Type III cell wall [40]. The synthesis of polysaccharides in cell walls also required an oxidized form of inositol [41]. The presence of glucuronic acid and mannose in the *C. delgadii* post-culture medium could indicate their involvement in the formation of the primary cell wall [42,43]. The presence of oligosaccharides (di- and trisaccharides) in the culture medium is probably connected with the senescence of cultivated cells, also the presence of reactive oxygen species (ROS) and the emergence of biotic or abiotic stresses during cell culture may influence cell senescence processes. Eusporangiate and leptosporangiate ferns have high concentrations of galacturonic acid and glucuronic acid in young, growing shoot tissue [42]. It is similar to the extracellular polysaccharides present in tobacco cell suspensions [44,45].

Sugar alcohols are acyclic polyols that play a key role in the metabolism of some higher plants. Their synthesis occurs in the photosynthesis system, and then, depending on the species, they are widely distributed in various plant organs. They belong to different groups according to their chemical structure and could be listed as: hydrogenated monosaccharides (sorbitol, mannitol), hydrogenated disaccharides (isomaltose, maltitol, lactitol) and mixtures of hydrogenated mono-di- and/or oligosaccharides (hydrogenated starch hydrolysate) [46]. Three of the sugar alcohols are widely distributed in angiosperms: galactitol, mannitol and sorbitol. Several studies suggest that sugar alcohols also play a role in tolerance to low temperature-, drought-, or salt-stress in plants. In the genetic manipulation of somatic cells, the sorbitol and the mannitol are used to build conditions of high osmotic pressure media for protoplasts isolation and their hybrid cultures [47]. For middle lamella construction, myo-inositol is delivered into the cell suspension culture by the organic elements of MS medium [48]. Sugar alcohols are primary photosynthetic products, which are also involved in the response of plant cell to stress by the change of the osmolarity of vacuole [49]. Out of 15 sugar alcohols listed in Appendix A, only four (i.e., maltitol, lactitol, sorbitol, xylitol) play an important role in the humane diet, reducing sugar content in the blood stream, and in modern nutrition [50].

Our analysis shows that the post-culture medium also contained several organic acids, as well as phosphoric acid. The latter was initially present as the salt KH_2_PO_4_, which is included in the basal MS medium at a concentration of 170 mg·L^−1^. The organic acids mentioned above are constituents of the metabolic pathways of some sugar polymers, such as pectins and lignins. Hexadecanoic acid occurs naturally in palm oil and is the most common saturated fatty acid. Phloroglucinol (a component of the flavonoid pathway), L-proline-5-oxo, N-acetyl-glucosoamine (an element of peptidoglucan involved in cell wall formation) and inositol, play a crucial role in the formation of the middle lamella. Chlorogenic acid isomers are a phenylpropanoid acid ester formed by the condensation of the carboxy group of trans-caffeic acid with the 3-hydroxy group of quinic acid. It is an intermediate metabolite in the biosynthesis of lignin [51]. On the level of scanning electron microscopic studies, it was shown that the dynamic growth of the cell suspension investigated here is characterized by the lack of secondary wall formation. Our previous studies of cell suspension of *Cyathea* provided evidence of a very thin middle lamella and only primary cell wall formation [5]. The cell walls of those cells located on the outside of the aggregates are distinctly ridged, which distinguishes them from cells of other cell suspensions, such as those of dicots like gentians [52]. 

The next less numerous groups of organic compounds recognized in post-culture medium of *C. delgadii* are fatty acids. In the plant world, the synthesis of fatty acids mainly occurs in chloroplasts from acetyl CoA, and generally relates to green tissue. Plant mitochondria are also capable of limited fatty acid biosynthesis [53]. We assume that in the case of our experimental subject, fatty acid biosynthesis occurs in the plastids, since the cell suspension is photosynthetically inactive. There is no photosynthesis, and it should be possible to combine with myo-inositol metabolism (100 mg/L in basic medium) or supported by a strong metabolism based on the enzymatic activity of dehydrogenase. The ketone or aldehyde groups are replaced by hydroxyl one. The analysis indicated several of the most common fatty acids, such as linolic acid, oleic acid, palmitic acid and stearic acid, that are part of cell membranes and belong to the unsaturated acid group. The majority of the listed fatty acids have C-chain without unsaturated bounds. The fatty acids facilitate the transport of hydrophilic natural products through cell organelle membranes; they are probably exuded from the cells to the medium together with more polar metabolites [54].

The confocal microscopy analysis of the cells of cultured tissue indicated the very reach metabolic activity of cytoplasm closely located to the cell wall, that might be the visualization of the place of flavonoid production [5]. It is very well known that flavonoids are widely distributed in the organs of growing ferns [55,56,57,58] and are even used for taxonomical purpose for species distinction in the frame of genus [59]. In cell suspension cells are getting old, and the exudation of flavonoids via the cell wall into the liquid medium was easily observed with the help of microscopic methods [5]. It resulted in yellow coloration of the medium. Our observation revealed that the intensity of coloration gradually increased due to the aging of a particular passage up to deep yellow. In the prolonged period of subculture up to 4–5 weeks, the cell proliferation stopped with an irreversible aggregation of the cells, the coloration of the whole culture turned from yellow to dark and finally turning black, the culture died. This is the result of cell starvation and most probably the degradation of flavonoids occurred, that resulted in the culture dying [5].

During the growing of plant material in subculture, an intensive yellow coloring of the liquid medium was observed. This fact directed our attention to the chemical composition of the post-culture medium. On the basis of exact *m*/*z* values of [M+H]^+^ or/and [M-H]^−^ ions and CID MS/MS spectra or comparison of the registered mass spectra with standards [60], six flavonoid derivatives were identified (Table 2). Mass spectrometric analysis revealed the presence of flavonols, flavanols and phenylpropanoid acid esters with quinic acid; it is supposed that among these compounds there might be compounds responsible for the yellow coloration, as many flavonoids are known to be the cause of color change of biological tissues/materials. Of course, the simple analysis performed for our needs does not prove the origin of the sample coloration, thus it must be considered as speculative until it is confirmed, e.g., by isolating the factor from the mixture. The identified flavonoids and chlorogenic acids may act as antioxidants limiting the damage caused by reactive oxygen species (ROS) during oxidative stress [36]. Additionally, three out of eight organic compounds listed in Table 2, namely, rutin, quercetin and chlorogenic acid, are characterized by pharmaceutical value and their potential activity against cancer [35]. The presence of these substances is very easily recognized in senescent cells that have decreased capacity for cell divisions. None rupture of the cell was observed and only exudation was the way for metabolites releasing into medium. The structural figure of the cell wall [5] was similar to that previously observed in wheat anther callus [61] and for the endosperm-derived callus of kiwifruit [62], that was supported by SEM analysis. So far, only the fern *Pteris vittata* has shown the presence of extracellular matrix during callus proliferation [63]. Having in mind the results presented in a previous paper [5] and the results presented here, especially the one concerning flavonoid bugs, the question of how they are exuded into surrounding liquid medium arises. The metabolite pathway of flavonoids is loosely bounded to the endoplasmic reticulum, and they are accumulated in the vacuole [64]. In cell aggregates of suspension cultures, the flavonoids are exuded into medium giving it a yellow coloration. The surplus chemicals are transferred from cell to cell via plasmodesmata (symplastic). Finally, they are secreted from plant cells as small endosomal derived membrane microvesicles (namely, exosomes) into extracellular space. The transmission scanning microscope enabled the observation of flavonoids on the outer side of cell walls that are permeable for natural products. The results presented here indicate that the organic substances secreted into the liquid medium are an expression of the intense metabolic activity of the cultured fern cells.

In conclusion, until now, little was known about the biological activity of cell suspension cultures in the tree ferns. The paper presents evidence of their unique experimental character. Analytical methods such as LC-MS and GC-MS revealed the presence of 11 metabolite groups with their preliminary identification based on retention times and registered mass spectra. Three of them: organic acids, sugars and sugar alcohols, were present in the largest number of identified natural products. Amino acids and fatty acids were the next numerous groups. The least numerous groups of metabolites were formed by alcohols, amines, phenolics, nucleic acids components and phosphates. Long-term culture provides the opportunity to conduct experiments involving the genetic manipulation of fern somatic cells based on protoplasts culture. In the future, manipulation of the medium composition and the application of various biotic and abiotic stress factors may allow to increase the efficiency of biosynthesis of specified secondary metabolites that show defined biological activity.

## 4. Material and Methods

### 4.1. Callus Induction and Cell Suspension Establishment

The callus induction and cell suspension establishment follow the methods earlier described for *Cyathea smithii* [5] and will be presented below. For callus induction, root explants were cultured on ½ MS agar medium [48] supplemented with 2,4-D (0.5; 1.0; 2.0 mg·L^−1^) and BAP (0.2 and 2.0 mg·L^−1^) in six combinations. For cell suspension establishment, yellow proliferating callus was transferred to ½ MS liquid medium supplemented with 2.0 mg·L^−1^ of 2,4-D and 0.2 mg·L^−1^ of BAP, and 20 g·L^−1^ sucrose (analytical grade). The establishment of suspension required an increment of the volume of the medium from 10, 20, 40, to 80 mL at two week long intervals. The suspension culture was produced by shaking at 120 rpm on a rotary horizontal shaker (Infors Rt 250, Switzerland) of 3 cm amplitude. The cell suspension was subcultured at two week intervals. All cultures were subjected to a photoperiod of 16/8 h day/night, at a temperature 22 ± 1 °C in a phytotron. All experiments were carried out on 3 years old, well established cell suspension.

### 4.2. Metabolic Profiling of the Post-Culture Liquid Medium Using LC-MS and GC-MS Analyzes

The 100 mL samples of liquid medium derived from a two week long passage of three years old suspension culture were centrifuged at 4000 rpm. Samples for LC-MS and GC-MS analyzes were prepared in triplicate. One mL of cell debris free, clear, transparent, yellow-colored supernatant was divided into two equal parts, the first was dried in vacuum centrifuge and resuspended after drying in 1 mL of 80% methanol. Then, the sample was placed in vial prior to analysis by LC-MS. For chromatographic separation, the Dionex RSLC 3000 was used. Separation was carried out on the C18 column Zorbax Eclipse XDB-C18 (Agilent) with a size of 2.1 × 100 mm with 1.8 μm bead size of resin. Separation was conducted at a 0.5 mL/min flow rate with two solvents creating the gradient: A (99.5% H_2_O/0.5% formic acid *v/v*) and B (99.5% acetonitrile/0.5% formic acid *v/v*). The tandem mass spectrometer Model timsTOF Pro (Bruker Daltonics, Bremen, Germany) was used, the system consisted of an electrospray ion source (ESI), and tandem mass analyzer with quadrupole and time of flight units (LC-ESI/MS/MS).

The ion source was operating at a voltage of 4 kV, nebulization was performed with nitrogen at 1.6 bar, dry gas flow was set up to 8.0 L min^−1^ and the source was heated to 220 °C. The instrument was maintained by the timsTOF Control ver. 2.3 and the acquired data were examined using the Bruker Data Analysis ver. 4.2. LC-MS profiles were registered in the positive and negative ion modes. As an internal standard an isoflavone genistein was used, its final concentration in the analyzed samples was 0.5 μg/μL. The amounts of identified compounds in plant tissues were expressed as relative values against the internal standard. The natural products were identified by comparison with their CID MS/MS spectra with those of standards, or present in mass spectral libraries [60].

For the analysis using GC-MS, the second part of each sample was dried under a stream of nitrogen and in a vacuum desiccator over P_2_O_5_. The compound derivatization in the sample was performed with 50 μL of methoxamine hydrochloride in pyridine (20 mg/mL) at 37 °C for 90 min with agitation. The second step of derivatization was performed by adding 100 μL of MSTFA (*N*-Methyl-*N*-trimethylsilyl)trifluoroacetamide) and incubation at 37 °C for 30 min with agitation. The samples were subjected to GC-MS analysis directly after derivatization.

GC-MS analysis was performed using TRACE 1310 gas chromatograph connected to the TSQ8000 triple-quad mass spectrometer (Thermo Scientific, Waltham, MA, USA). A DB-5MS bonded-phase fused-silica capillary column (30 m length, 0.25 mm inner diameter, 0.25 μm film thickness) (J&W Scientific Co., Folsom, CA, USA) was used for separation. The GC oven temperature gradient was as follows: 70 °C for 2 min, followed by 10 °C/min up to 300 °C (10 min), 2 min at 70 °C, raised by 8 °C/min to 300 °C and held for 16 min at 300 °C. For sample injection, a PTV injector was used in a range of 60 to 250 °C, transfer line temperature was set to 250 °C, and source to 250 °C. The spectra were recorded in *m*/*z* range of 50–850 in EI+ mode with an electron energy of 70 eV. Raw MS-data were analysed using the MSDial 4.0 software package. To eliminate retention time (Rt) shift and to determine the retention indexes (RI) for each compound, the alkane series mixture (C-10 to C-36) was injected into the GC-MS system. The identified artifacts (alkanes, column bleed, plasticizers, MSTFA, and reagents) were excluded from further analyses.

## Figures and Tables

**Figure 1 ijms-23-11783-f001:**
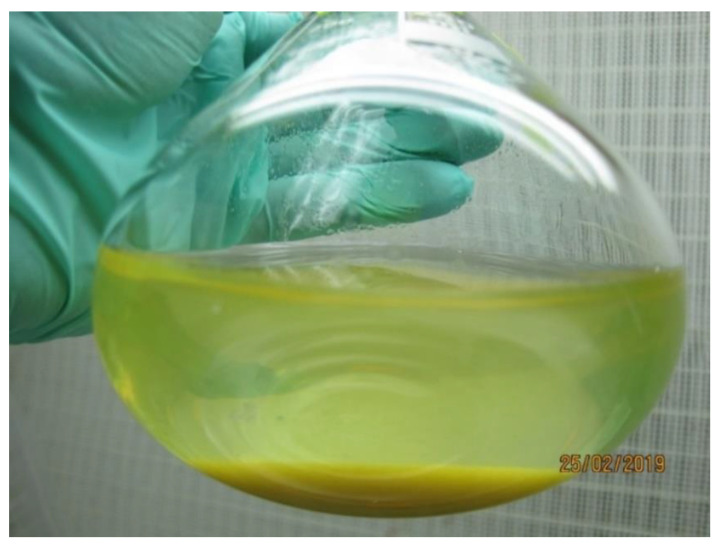
Conical flask with 3 years old cell suspension culture of *Cyathea delgadii* with yellow coloration of tissue and liquid medium.

**Figure 2 ijms-23-11783-f002:**
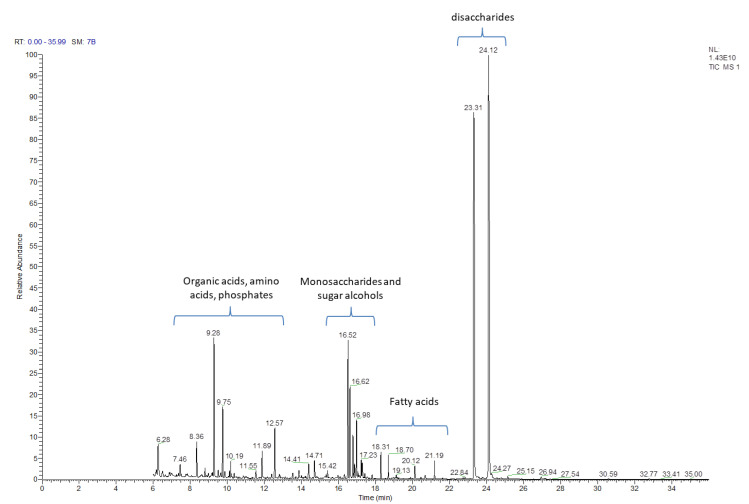
GC-MS chromatogram of post-culture medium after two week long culture of the *Cyathea delgadii* cell suspension.

**Figure 3 ijms-23-11783-f003:**
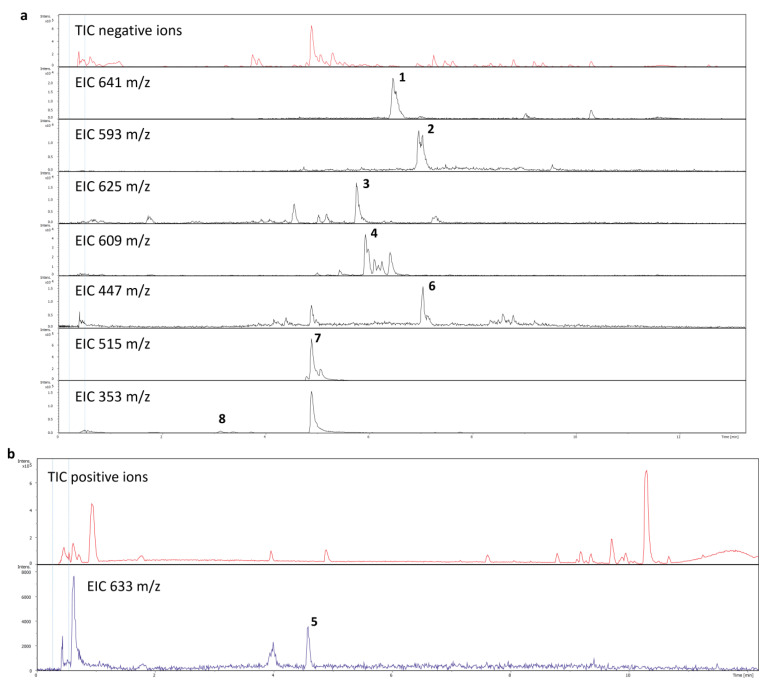
Total ion and extracted ion chromatograms of post culture medium registered in negative (**a**) and positive (**b**) MS mode.

**Figure 4 ijms-23-11783-f004:**
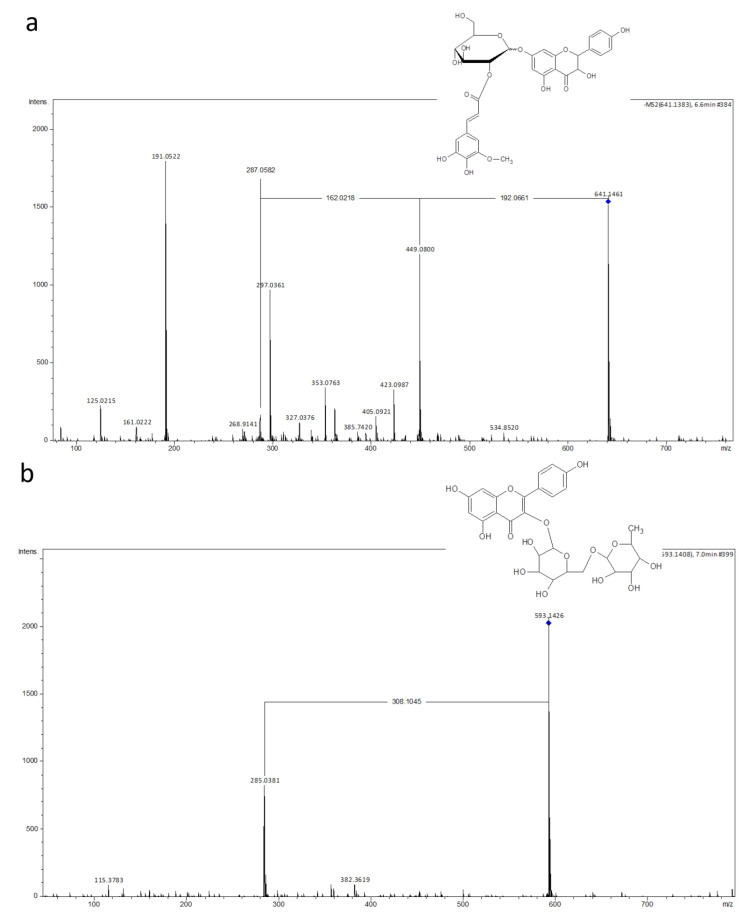
Spectra of two selected flavonoids identifed after LC-MS analysis of post culture medium of *Cyathea delgadii*: (**a**) Aromadendrin 7-O-(hydroxyferuloyl-glucoside); (**b**) Kaempferol 3-O-rutinoside (rhamnosyl-glucoside).

**Table 1 ijms-23-11783-t001:** The range of retention time of alphabetic list of groups of organic chemicals recognized in post culture liquid medium of *Cyathea delgadii* cell suspension culture.

Number	Class of Metabolites	Number of Metabolites	Retention Time [min]	
			Minimal Value	Maximal Value
1.	Alcohols	4	9.323	17.473
2.	Amines	4	6.976	19.088
3.	Amino acids	10	7.344	15.272
4.	Fatty acids	12	7.854	21.806
5.	Nucleic acids components	5	7.456	12.582
6.	Organic acids	29	6.286	24.531
7.	Phenolics	6	11.850	19.143
8.	Phosphates	6	6.653	21.541
9.	Sugars	27	11.708	32.775
10.	Sugar alcohols	15	12.381	25.146
11.	Others	12	6.038	19.556

**Table 2 ijms-23-11783-t002:** Flavonoids identified with LC-MS. List of flavonoids, including polyphenol and their biological activity identified in post-culture medium of the *Cyathea delgadii* cell suspension.

No.	Compound	Polarity Ion Mode	Molecular Formula	Obtained *m/z*	Calculated *m/z*	Error [ppm]	Biological Role /Activity ^c^
1	aromadendrin 7-O- (hydroxyferuloyl- glucoside) ^a^	-	C_31_H_30_O_15_	641.1461	641.1512	−4.6	Anti-inflammatory, antioxidant, antidiabetic, ROS scavenger
2	kaempferol 3-O-rutinoside (rhamnosyl-glucoside) ^a^	-	C_27_H_29_O_15_	593.1426	593.1512	−2.55	In plants ROS scavenger
3	aromadendrin 7-O- (hydroxyferuloyl- rhamnoside) ^a^	-	C_31_H_30_O_14_	625.1518	625.1562	−3.71	Anti-HIV-1 activity
4	naringenin 7-O- (hydroxyferuloyl- rhamnoside) ^a^	-	C_31_H_30_O_13_	609.1626	609.1613	−2.0	Anti-inflammatory properties, antifibrogenic effects
5	quercetin 3-O-rutinoside–(rhamnosyl-glucoside) ^b^	+	C_31_H_30_O_13_	633.1967	633.1426	4.47	Reduction of the levels of oxidative stress in the colon, ROS scavenger, pharmacological benefits for the treatment of various chronic diseases such as cancer, diabetes, hypertension and hypercholesterolemia
6	quercetin rhamnoside	-	C_27_H_30_O_16_ [sodiated ion]	447.1200	447.0932	3.08	Cytotoxic effect on breast cancer, ROS scavenger
7	1,5-dicaffeoylquinic acid (Cynarin) ^a^	-	C_25_H_24_O_12_	515.1098	515.1195	2.07	Protective role in the control of oxidative damage, ROS scavenger
8	chlorogenic acid ^b^	-	C_16_H_18_O_9_	353.0859	353.0878	1.12	Anti-cancer activity, functioning as an intermediate in lignin biosynthesis

^a^ identification based on exact molecular mass, error of *m/z* value for protonated molecule [M + H]^+^ below 5 ppm and registered CID MS/MS spectra; ^b^ identified after comparison with mass spectrum of standards; ^c^ PubChem data base and references: no. 1. [32]; no. 2. [33]; no. 3. [34]; no. 4. https://en.wikipedia.org/wiki/Naringin; no. 5. [35]; no. 6. https://en.wikipedia.org/wiki/Quercetin; no. 7. [36]; no. 8. https://en.wikipedia.org/wiki/Chlorogenic_acid (accessed on 1 July 2022).

## Data Availability

The original contributions presented in the study are included in the article/Supplementary Material. Further inquiries can be directed to the corresponding author.

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
