# Peer review of "The Metabolite Content of the Post-Culture Medium of the Tree Fern Cyathea delgadii Sternb. Cell Suspension Cultured in the Presence of 2,4-D and BAP"

_ijms, 2022, doi:10.3390/ijms231911783_

Round 1
Reviewer 1 Report
In this research, GC/MS and LC/MS were used to analyze the metabolite content of the post-culture medium of Cyathea delgadii cell suspension in the presences of 2,4-D and BAP. Although this work solves some interesting questions, such as identifying some metabolite groups of cell suspension cultures, overall, the study does not point to the biological significance of these metabolites to Cyathea delgadii.
1.The purpose of this paper is to analyze the primary and secondary metabolites in the post-culture liquid medium of Cyathea delgadii cell suspension, but the primary and secondary metabolites are not discussed and analyzed respectively in the results.
2.No LC/MS chromatograms is attached to the paper, and there is no clear explanation for the LC/MS results.
In general, the results obtained in this paper are few, and the conclusion are not in-depth enough. Only a preliminary analysis is carried out, and the biological significance of these metabolites to Cyathea delgadii is not pointed out.
Minor Concerns: 1. The fonts in line 149 to line 154 are inconsistent with other fonts.
2. Each reference has two serial numbers.
3. Authors may need to un-underline the authors of references 30, 45 and 50.
Author Response
Answer for Comments and Suggestion of Reviewer 1.
1.The purpose of this paper is to analyze the primary and secondary metabolites in the post-culture liquid medium of Cyathea delgadii cell suspension, but the primary and secondary metabolites are not discussed and analyzed respectively in the results.
Just as the eastern hemisphere species of Cythea genus have been the objective of interest of autochthons in the past and at present of the science (Cyathea australis, C, smithii), so in case of the western hemisphere for species of Cyathea genus specially C. delgadii there is no such evidences. Because of ours activity the species C. delgadii had been discovered for science as the objective of experimental botany and biotechnology (first time described somatic embryogenesis for ferns and very high morphogenic potential of isolated and cultured apical dome of in vitro cultured sporophytes). And carry on discussion on present matter meets problems, which depends on the fact of the lack of literature concerns the tree ferns.
Supplementary data includes Table containing the list of all found organics (130) and Tab. 1 shows their groups. Each large group among them have been discussed. The second part of discussion is devoted to flavonoids presented in Tab. 2.
- No LC/MS chromatograms is attached to the paper, and there is no clear explanation for the LC/MS results.
LC/MS results have been supplemented by total ion and extracted ion chromatograms
The paragraph “In conclusion” was modified underlying the exploration of present data for future experiments.
Minor Concerns:
- The fonts in line 149 to line 154 are inconsistent with other fonts.
- Each reference has two serial numbers.
- Authors may need to un-underline the authors of references 30, 45 and 50.
All three mentioned comments were considered.
Reviewer 2 Report
I have only one comment to ask to the authors: does the analyzed samples belong, as it seems, to the same subculture? If this is the case, in my opinion, they should mention about possible effects of successive subculturing on the stability of the production of the observed metabolites
The numbering of the bibliography is repeated twice.
Author Response
Answer for comments of reviewer 2
All analysis has been done on one subculture of three-year-old well established cell suspension of C. delgadii. Founding on structural and physiological levels help us to select stage of cell suspension growth (Rybczyński et al. 2022).
Round 2
Reviewer 1 Report
this manuscript "The metabolite content of the post-culture medium of the tree 2 fern Cyathea delgadii Sternb. cell suspension cultured in the 3 presence of 2,4-D and BAP" is suitable for publication.
Author Response
Thank you.